# Antifungal Activity of an Original Amino-Isocyanonaphthalene (ICAN) Compound Family: Promising Broad Spectrum Antifungals

**DOI:** 10.3390/molecules25040903

**Published:** 2020-02-18

**Authors:** Miklós Nagy, Gábor Szemán-Nagy, Alexandra Kiss, Zsolt László Nagy, László Tálas, Dávid Rácz, László Majoros, Zoltán Tóth, Zsuzsa Máthéné Szigeti, István Pócsi, Sándor Kéki

**Affiliations:** 1Department of Applied Chemistry, Faculty of Science, University of Debrecen, 4010 Debrecen, Hungary; miklos.nagy@science.unideb.hu (M.N.); sporadicprion@gmail.com (Z.L.N.); jenyiszjin@gmail.com (D.R.); 2Department of Molecular Biotechnology and Microbiology, Faculty of Science, University of Debrecen, 4010 Debrecen, Hungary; szeman-nagy.gabor@science.unideb.hu (G.S.-N.); kissalexandra0329@gmail.com (A.K.); talas.laszlo@science.unideb.hu (L.T.); szigeti.zsuzsa@science.unideb.hu (Z.M.S.); 3Department of Medical Microbiology, Faculty of Medicine, University of Debrecen, 1 Egyetem tér, 4010 Debrecen, Hungary; major@med.unideb.hu (L.M.); toth.zoltan@med.unideb.hu (Z.T.)

**Keywords:** multiple drug resistance, antifungal effect, 1-amino-5-isocyanonaphthalenes, 1,1-*N*-dimethylamino-5-isocyanonaphthalene, *Candida* species, *C. albicans*

## Abstract

Multiple drug resistant fungi pose a serious threat to human health, therefore the development of completely new antimycotics is of paramount importance. The in vitro antifungal activity of the original, 1-amino-5-isocyanonaphthalenes (ICANs) was evaluated against reference strains of clinically important *Candida* species. Structure-activity studies revealed that the naphthalene core and the isocyano- together with the amino moieties are all necessary to exert antifungal activity. 1,1-*N*-dimethylamino-5-isocyanonaphthalene (DIMICAN), the most promising candidate, was tested further in vitro against clinical isolates of *Candida* species, yielding a minimum inhibitory concentration (MIC) of 0.04–1.25 µg/mL. DIMICAN was found to be effective against intrinsically fluconazole resistant *Candida krusei* isolates, too. In vivo experiments were performed in a severly neutropenic murine model inoculated with a clinical strain of *Candida albicans*. Daily administration of 5 mg/kg DIMICAN intraperitoneally resulted in 80% survival even at day 13, whereas 100% of the control group died within six days. Based on these results, ICANs may become an effective clinical lead compound family against fungal pathogens.

## 1. Introduction

Invasive fungal infections cause 1.5 million deaths annually and may show a further increase in the following decades. Most of these infections are caused by *Candida* species, followed by *Aspergillus* and *Cryptococcus neoformans* [1,2,3].

Members of the *Candida* genus are part of the healthy human microbiome and can be found in the oral cavity, gastrointestinal and urogenital tract, however as opportunistic pathogens, these yeasts can cause a wide variety of diseases ranging from superficial infections to life threatening invasive candidiasis [4,5,6]. Incidence and mortality of the latter remained high in the last decades, despite the widespread use of echinocandins and newer generation triazoles as prophylactic and therapeutic agents [7,8]. The development of invasive candidiasis is associated with several predisposing factors, notably with immunosuppression, recent abdominal surgery, diabetes, broad spectrum antibiotic therapy and many others [9]. While risk factors are numerous, therapeutic options are very limited with only three major antifungal classes (triazoles, polyenes, echinocandins) available [10]. The most recent antifungal compounds, the echinocandins were introduced almost twenty years ago and there are few new drugs in the pipeline in the following years [11]. It should also be noted that several strains of the *Candida* species, such as *Candida krusei* exhibit intrinsic resistance to fluconazole, however despite their potential to emerge as multidrug-resistant pathogens in the hospital setting, this has not yet occurred [1]. Nevertheless, the emergence of new, drug resistant pathogenic fungi, such as *Candida auris* pose a serious therapeutic challenge and highlights the need for new compounds with different mechanisms of action [12,13].

Recently, we developed a novel amino-isocyanonaphthalene (ICAN) based solvatochromic fluorophore family [14], which despite their very simple structure and easy preparation, still mark a white spot on the map of chemistry. Despite their relative novelty, they already found numerous and versatile use in both analytical chemistry [15,16,17] and cell-biology [18,19,20]. During our experiments to utilize ICAN derivatives as vital stains on CaCo2, OCM-1, HuLi and HaCat cell lines [20] we noticed that they are perfectly suitable for the staining of different fungi, too. However, after testing 1-*N*-methylamino-5-isocyanonaphthalene (MICAN) on a commercial strain of *Saccharomyces cerevisiae*, a strong antimycotic effect was observed. We wondered that ICAN derivatives could be used as potential drugs against pathological fungi, and planned experiments on *Candida* species were started.

The aims of this study were to test the antifungal activity of 1-amino-5-isocyanonaphthalene and its derivatives on different *Candida* species in vitro and to test the most effective agent in vivo, in a murine model of invasive candidiasis. This discovery can lead to the development of a new original compound family, which can rival the currently approved drug classes or even top them in several fields of application. It should be noted, however that this study focuses only on the 1,5-ICAN derivatives, whereas ICANs are easy to modify and even the slightest change in the relative substitution position of the amino and isonitrile groups on the naphthalene ring can result in a completely different behavior [21].

## 2. Results

### 2.1. Antifungal Activity of the ICAN Derivatives

We previously carried out extensive studies to successfully utilize 1-*N*-methylamino-5-isocyanonaphthalene (MICAN) as a vital cell dye on different human cell lines [20]. The compound along with ICAN and 1,1-*N*-dimethylamino-5-isocyanonaphthalene (DIMICAN) are well tolerated by human cell cultures, their LD_50_ values (on HaCat cell line) were determined (MTT bio assay technique) to be 10 μg/mL for ICAN, 18 μg/mL for MICAN, and 13 μg/mL for DIMICAN. Encouraged by these results we wanted to extend their applicability to fungal staining. Since these dyes are nonpolar, they are expected to easily bind to and stain the nonpolar chitin in the cell wall of different fungal species. We tested this assumption on common yeast from a local shop, and much to our surprise after staining, the yeast cells stopped growing and died in a short time. Repeated experiments led to the same conclusion; however, the applied dye concentration was well below the LD_50_ determined for human cells. We assumed that these compounds may have antifungal effect, therefore *Candida albicans*, one of the most common human pathogenic fungi was selected for further tests. HaCat cell cultures were infected with *C. albicans* and were treated with different concentrations of MICAN dissolved in DMSO. The fungal growth was followed by time lapse imaging for 24 h. The results are summarized in Figure 1. As it is evident from Figure 1a,b, contrary to the untreated (DMSO control) cells, which show the typical yeast growth curve (Figure 1b), little fungal growth (~30 % of the starting *C. albicans* cells even germinated) was observed at even as low as 7.5 μg/mL MICAN concentrations, which is well below its LD_50_ value on HaCat cells. It should be noted, however, that in the case of the untreated coculture, the fungal growth exceeded 100% of the field-of-view (24 h) developing three-dimensional, multilayered hyphal mass. Additionally, the average hyphal area was 10-fold higher than in the case of the lowest MICAN concentration applied. The antifungal effect of MICAN (and DIMICAN) is also presented in the Appendix A. Despite effective fungal growth inhibitory of MICAN, the treated HaCat cells showed no sign of damage under 24 h.

### 2.2. Antifungal Susceptibility Testing–Determination of Structure-Activity Relationship

ICAN has 3 key moieties, namely the amino, naphthalene and isonitrile. To find out which ones of them are essential for the antifungal behavior and to design more efficient derivatives, comparative studies were carried out, where one of the moieties was exchanged or eliminated. The modification/synthetic strategy is summarized in Figure 2. All ICAN derivatives were tested against *C. albicans* SC5314, *C. krusei* ATCC 6258 and *Candida parapsilosis* ATCC 22019. Table 1 summarizes the minimum inhibitory concentrations (MICs) of the ICAN derivatives. The antifungal agents were dissolved in 100% DMSO and diluted further in RPMI-1640 (with L-glutamine but without bicarbonate) and buffered to pH 7.0 with 0.165 M morpholinopropanesulfonic acid (MOPS). MICs were read visually after 48 h incubation at 35 °C using the partial (50%) inhibition (for ICAN derivatives) (PI) and total inhibition (for amphotericin B (AMB) and ICAN derivatives) (TI) criteria. MICs were determined at least twice.

Based on the data of Table 1 it can be concluded that even the simplest amino-isocyanonaphthalene molecule ICAN has good antifungal effect with MIC_TI_ values between 0.6–5 µg/mL. The starting compound 1,5-diaminonaphthalene (DAN) proved to be completely ineffective against *Candida* (MIC_TI_ > 100 µg/mL), therefore we can conclude that the isonitrile group is essential for the antifungal effect. Next the free amino group of ICAN was converted to isonitrile resulting 1,5-diisocyanonaphthalene (DIN), which showed similar (or higher) antifungal activity (MIC_TI_ = 0.6 µg/mL) compared to that of ICAN. However, the largely nonpolar and rigid aromatic structure of DIN significantly limits its solubility in polar solvents, such as water. We previously also encountered solubility problems with DIN, when we tried to determine its fluorescent properties [14]. Nevertheless, DIN may also become an effective antifungal drug after solving the solubility issues with proper formulation. The naphthalene core of DIN was exchanged to phenylene to obtain 1,4-phenylenediisocyanide (PDI). In contrast to DIN, PDI proved to be completely ineffective (MIC_TI_ > 100µg/mL), which means that the naphthalene core is also essential for the antifungal behavior (Figure 2.). To increase the efficacy of ICAN derivatives, the substitution of the amino group followed. The alkylation of the amino group increases the dipolarity of the molecules, however at the same time it reduces their ability to form H-bonding as an H-bond donor. Monomethylation (MICAN) led to lowered MIC_TI_ values (0.75–1.25 µg/mL), however the use of longer alkyl chains, such as ethyl and propyl (EICAN and PICAN) proved to be not so effective (MIC_TI_ = 1.25–2.5 µg/mL and 1.75–2.5 µg/mL, respectively), maybe due to steric hinderance. The lowest MIC values were obtained in the case of DIMICAN (MIC_PI_ ≤ 0.04 µg/mL and MIC_TI_ = 0.3–0.6 µg/mL), the dimethylated ICAN derivative. It should be noted that the MICs did not vary significantly for most of the compounds under investigation, only in the case of ICAN was a broader range observed (9-fold variation for ICAN (MIC = 0.6–5 µg/mL). For further tests we chose one of the most effective ICAN derivative, DIMICAN which has no cytotoxic effect [15,20]. To test the “real life” behavior of DIMICAN it was tested for in vitro susceptibility against clinical isolates collected from patients at the Clinics of the University of Debrecen. Table 2 summarizes the MICs of the new antifungal agent DIMICAN.

Amphotericin B susceptibility of the isolates was also tested, and their MIC_TI_ values were not higher than 2 µg/mL in each case. DIMICAN MIC_PI_ values were same or significantly lower compared to MIC_TI_ for all tested five *Candida* species. The lowest MICs were noticed in case of *C. albicans*.

### 2.3. Mouse Model of Infection and Antifungal Therapy

In the model of acute invasive candidiasis, neutropenic mice were infected via the lateral tail vein with *C. albicans* 3666 isolate (3 × 10^4^ cfu/ mouse) to establish an acute infection. All untreated control animals died within six days. DIMICAN (5 mg/kg) prolonged the survival period of animals compared to non-treated infected controls (*p* < 0.05; 80 versus 0% survivors at day 7 post-infection; Figure 3a). It should be noted, however, that DIMICAN was administered as intraperitoneal injection which contained DIMICAN as an emulsion in reagent grade olive oil and water. The effect of the pure vehicle (oil/water emulsion) was tested on eight-week-old mice and aside from increased weight gain compared to the control group, no mortality was observed within five days. Proper clinical formulation may have resulted in better survival of the treated group, however it is a complex task for this new molecule and therefore exceeds the scope of this paper.

### 2.4. Determination of Fungal Burden

During an independent in vivo experiment, mice were infected intravenously with *C. albicans* 3666 isolate. The infected mice were euthanized two or six days after infection (*n* = 5 and 6 mice/time point, respectively) to determine the fungal burden in the kidney. Kidney burden was analyzed using a Kruskal–Wallis test with Dunn’s posttest for multiple comparisons [23,24]. Results of fungal burden experiments are shown on Figs. 3b and c. Two days after infection AMB and DIMICAN had no effect on fungal load (*p* > 0.05). Six days after infection AMB and DIMICAN treatment resulted in significantly lower fungal burden compared to control: *p* < 0.05 (AMB); *p* < 0.001 (DIMICAN). It is evident in Figure 3a,b, that on day six the fungal burden increased by an order of magnitude (from ~10^5^ to ~10^6^) in the case of the untreated mice, while in the case of DIMICAN it remained the same as it was on day two. The same is valid for AMB. Based on these results it can be concluded that DIMICAN is at least as effective in reducing the fungal burden in the kidney as amphotericin B.

## 3. Discussion

It was discovered that our new, original molecule: 1-amino-5-isocyanonaphthalene (ICAN) and its derivatives exert excellent antifungal activity against a broad range of *Candida* species. Structure-activity studies based on minimum in vitro inhibitory concentration (MIC) determined for three *Candida* species, namely *C. albicans*, *C. krusei* and *C. parapsilosis* revealed that for the antifungal effect the naphthalene core, the isocyano- and the amino-moieties are all necessary. It was noted that alkylation especially methylation considerably increased the antifungal activity. However, 1,5-diisocyanonaphthalene (DIN) also showed low MIC (MIC_PI_ = 0.3 μg/mL, MIC_TI_ = 0.6 μg/mL), but its rigid structure and consequently its low solubility in aqueous media may limit its application. Nevertheless, proper formulation and/or substitution with solubility enhancer groups on the naphthalene core may increase its effectiveness. Based on MIC values (MIC_PI_ = 0.04–0.3 μg/mL, MIC_TI_ = 0.6 μg/mL) and its low cytotoxicity 1,1-N-dimethylamino-5-isocyanonaphthalene (DIMICAN) was selected to be tested on clinical *Candida* isolates. In vitro we tested it using microdilution method against *Candida albicans* (8 isolates), *C. krusei* (7 isolates), *C. tropicalis* (7 isolates), *C. glabrata* (7 isolates) and *C. parapsilosis* (5 isolates) to get total minimum inhibitory concentration (MIC_TI_) values of 0.08–1.25 µg/mL. These MIC values are comparable or lower than that of amphotericin B for the same isolates. It is important to note that DIMICAN was found to be effective against intrinsically fluconazole resistant *C. krusei* isolates, too. The in vivo applicability of DIMICAN was tested on immunosuppressed mice and control experiments were also conducted with amphotericin B. In the model of acute invasive candidiasis, neutropenic mice were infected intravenously with *C. albicans* 3666 isolate (3 × 10^4^ cfu/ mouse) to establish an acute infection, and all untreated control animals died within six days. DIMICAN (5 mg/kg) prolonged the survival period of animals compared with the survival period of non-treated infected controls, *P* < 0.05; 80 versus 0% survivors at day 7 post infection. Results of fungal burden in kidney revealed that six days after infection AMB and DIMICAN were efficient against *Candida* isolate: *p* < 0.05 (AMB); *p* < 0.001 (DIMICAN). Based on these results and the easy and versatile modification of the ICANs we hope that they have the potential to become an effective clinical lead compound family against pathogenic fungi. As potential lead compounds, ICANs, meet one or more criteria of commercial use, such as low cost, easy preparation and good chemical stability, for example after extensive toxicity and time-kill studies may be used as antimycotic coatings on medical tubes and/or devices. This simple, cheap, nontoxic and easy to prepare compound family would allow not only human treatment, but at the same time could be used as disinfectants and/or coating which is vital against the spread of highly virulent Candida strains.

## 4. Materials and Methods

### 4.1. Materials

1,5-diaminonaphthalene (DAN, Aldrich, D21200) was used as received. The synthesis of the ligands: 1-amino-5-isocyanonaphthalene (ICAN), 1-N-methylamino-5-isocyanonaphthalene (MICAN) 1-*N*,*N*-dimethylamino-5-isocyanonaphthalene (DIMICAN) and 1,5-diisocyanonaphthalene (DIN) is found in refs.: 14 and 17.

### 4.2. General Method for the Synthesis of Alkylated 1-amino-5-isocyanonaphthalenes

A 250 mL round-bottomed flask, equipped with a magnetic stir bar was charged with 1-amino-5-isocyanonaphthalene (1.00 g, 5.94 mmol), potassium hydroxide (0.670 g, 11.9 mmol) and absolute dimethyl formamide freshly distilled over phosphorus pentoxide (100 mL). Alkyliodide (59.4 mmol)* was added to the solution, then the flask was flushed with argon and sealed with a rubber septum. The reaction mixture was stirred at room temperature, protected from light. After two days 200 mL methylene chloride and 5% ammonia solution was added, and the solution was extracted 5 times with water, then the organic phase was dried over anhydrous magnesium sulfate. Solvent was removed on a rotary evaporator and the residue was purified on a normal-phase silicagel filled column, using methylene chloride: hexane (1:1) as eluent.

Ethyliodide (Sigma-Aldrich, Schnelldorf, Germany) was used to synthesize 1-ethyl-5-isocyanonaphthalene (EICAN) (yellow crystals, 370 mg, 32% yield) and diEICAN (light green waxy compound, 90 mg, 6.8% yield). To synthesize1-propyl-5-isocyanonaphthalene (PICAN) propyliodide (Sigma-Aldrich, Germany) was used. PICAN (light yellow waxy compound, 440 mg, 35% yield) and diPICAN (green waxy compound, 100 mg, 6.7% yield). With the excess of alkyl iodide used the ratio of the mono- and dialkylated-product formed can be controlled.

*EICAN (1-N-ethylamino-5-isocyanonaphthalene)*^1^H NMR (400 MHz, CDCl_3_) δ = 7.83 (d, *J* = 8.6 Hz, 1H), 7.50 (q, *J* = 8.6 Hz, 3H), 7.32 (dd, *J* = 8.5, 7.5 Hz, 1H), 6.66 (dd, *J* = 6.8, 1.6 Hz, 1H), 4.30 (s, 1H), 3.29 (q, *J* = 7.1 Hz, 2H), 1.39 (t, *J* = 7.1 Hz, 3H) ppm. ^13^C NMR (101 MHz, CDCl_3_) δ = 166.70 (quart. Ar-NC), 144.05 (quart. Ar), 129.06 (Ar), 124.49 (Ar), 123.52 (quart. Ar), 123.21 (Ar), 121.85 (Ar), 111.48 (Ar), 105.55 (Ar), 38.69 (Ethyl–CH_2_–), 14.65 (Ethyl–CH_3_) ppm.

*PICAN (1-N-propylamino-5-isocyanonaphthalene)*^1^H NMR (400 MHz, CDCl_3_) δ = 7.83 (d, *J* = 8.6 Hz, 1H), 7.50 (q, *J* = 8.6 Hz, 3H),7.32 (dd, *J* = 8.5, 7.5 Hz, 1H), 6.66 (dd, *J* = 6.4, 1.8 Hz, 1H), 4.38 (s, 1H), 3.22 (t, *J* = 7.0 Hz, 2H), 1.85–1.72 (m, 2H), 1.08 (t, *J* = 7.4 Hz, 3H) ppm. ^13^C NMR (101 MHz, CDCl_3_) δ= 166.70 (quart. Ar-NC), 144.10 (quart. Ar), 129.06 (Ar), 124.49 (Ar), 123.53 (quart. Ar), 123.19 (Ar), 121.81 (Ar), 111.36 (Ar), 105.50 (Ar), 45.96 (Propyl–CH_2_–), 22.51 (Propyl–CH_2_–), 11.81 (Propyl–CH_3_) ppm.

### 4.3. Time-lapse Microscopy and Digital Image Analysis of Cocultures of HaCaT cells and C. albicans

Human keratinocytes were grown under standard cell breeding conditions in T-25 culture vessels. At 40% of cellular monolayer confluency the cultures were treated and observed under four parallel custom-built inverted microscopes housed in a cell-breeding incubator. 1.5 × 10^5^/mL *C. albicans* cells were used for inoculation. Image acquisitions were carried out at 1 frame/min frequency. Exposure-synchronized low-energy near-infrared illumination at 940 nm was used for the sake of minimal invasivity. Image sequences were preprocessed prior analysis to eliminate flickering and uneven illumination artefacts. Quantitative image analysis were done via NIH (US National Institutes of Health) ImageJ v1.39d software bundle, using hyphal separation plugins. Determination of hyphal area was preferred for precision against length, because of the frequent overlapping and ramifications in control samples.

### 4.4. Clinical Isolates and Antifungal Susceptibility Testing

The tested *C. albicans*, *C. tropicalis*, *C. parapsilosis*, *C. glabrata* and *C. krusei* ATCC strains and clinical isolates (collected from sterile body sites) are listed in Table 2. MICs were determined according to the CLSI M27-A3 [22]. Antifungal agents were dissolved in 100% DMSO and diluted further in RPMI-1640 (with L-glutamine but without bicarbonate) (Sigma, Budapest, Hungary), and buffered to pH 7.0 with 0.165 M morpholinopropanesulfonic acid (MOPS; Sigma, Budapest, Hungary). The final concentration ranges used for AMB and ICAN derivatives were 0.125 to 4 µg/mL and 0.04 to 20 µg/mL, respectively. Cell suspensions were prepared in RPMI 1640 medium and were adjusted to give a final inoculum concentration of 0.5 × 10^3^ to 2.5 × 10^3^ cells/mL. MICs were read visually after 48-h incubation at 35 °C using the partial (50%) inhibition (for ICAN derivatives) (PI) and total inhibition (for AMB and ICAN derivatives) (TI) criteria. MICs were determined at least twice. For amphotericin B isolates with MIC values of ≤2 mg/L were considered wild-types [25].

### 4.5. Mouse Model of Infection

We used 10–12 week-old female BALB/c mice for all experiments. Mice were immunosuppressed using 150 mg/kg intraperitoneal (ip) cyclophosphamide dose (4 days prior to infection) and 100 mg/kg ip cyclophosphamide dose (1 day prior to infection; 2 and 5 days after infection) [26]. All mice received ceftazidime (5 mg/day subcutaneously) from days 0 to 6 after infection. Mice were given food and water ad libitum and were monitored daily. The animals were maintained in accordance with the Guidelines for the Care and Use of Laboratory Animals, and the experiments were approved by the Animal Care Committee of the University of Debrecen, Debrecen, Hungary (permission no. 12/2014).

Mice were infected via the lateral tail vein with *C. albicans* 3666 bloodstream isolate. In the lethality (10 mice/group) and fungal tissue burden experiments (6 mice/group) the infectious doses were 3 × 10^4^ and 10^4^ cfu/ mouse, respectively. These doses led to 100% and 0% mortality, respectively by day 6 after infection among untreated mice. Inoculum density was confirmed by plating serial dilutions onto Sabouraud dextrose agar (SDA) plates.

### 4.6. Antifungal Therapy

AMB (Fungizone, commercial preparation) and DIMICAN treatment in a 0.5-mL bolus was started 1 h after infection. The doses were 1 mg AMB and 5 mg DIMICAN (in olive oil/water (o/w) emulsion) per kg body weight. All antifungals were administered intraperitoneally once daily for 5 consecutive days.

### 4.7. Determination of Fungal Burden

Infected mice were euthanized 2 and 6 days after infection (*n* = 6 mice/time point) to determine the fungal burden in the kidney. The organs were aseptically removed, weighed, homogenized in sterile saline, serially diluted 10-fold in sterile saline. Aliquots of 0.1 mL of the undiluted and diluted homogenates were plated onto SDA, and the colony count was determined after 48 h. Colony-forming units (CFUs) were determined after 48 h of incubation at 37 ° C and results were expressed as cfus/g tissue [26]. The lower limit of detection was 50 cfu/g tissue. Kidney burden was analyzed using a Kruskal–Wallis test with Dunn’s posttest for multiple comparisons [23,24].

### 4.8. Statistical Test

Survival effects were analyzed by the Kaplan–Meier log-rank test. Kidney burden was analyzed using a Kruskal–Wallis test with Dunn’s posttest for multiple comparisons [23,24].

## Figures and Tables

**Figure 1 molecules-25-00903-f001:**
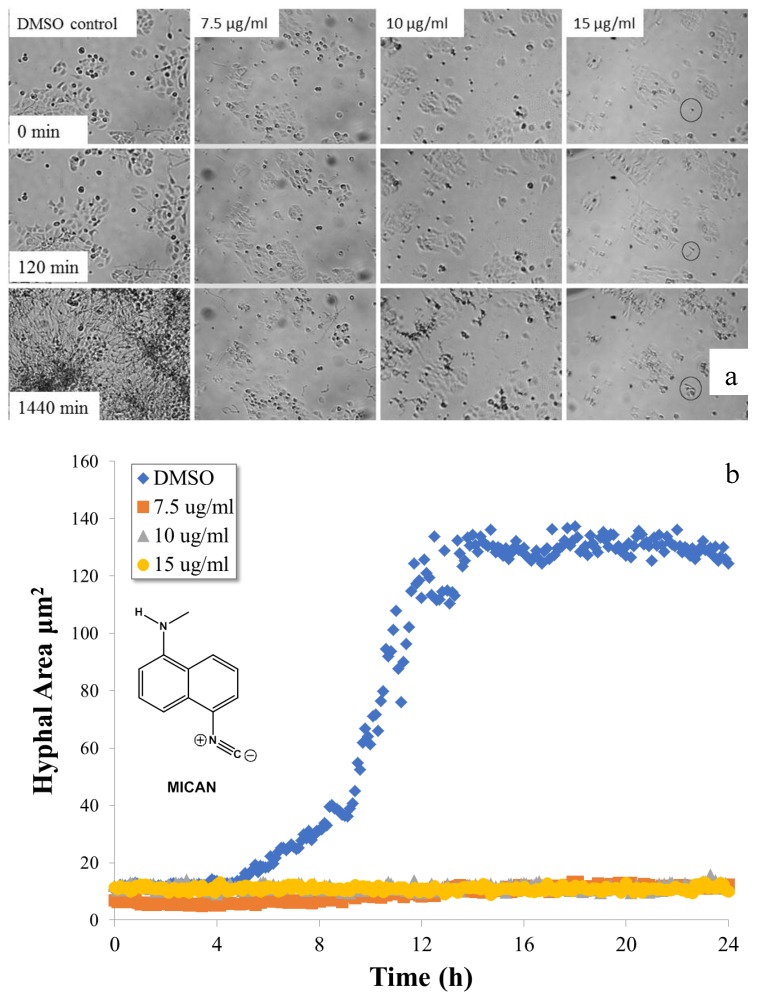
Hyphal growth of *C. albicans* in the presence of methylamino-5-isocyanonaphthalene (MICAN). Time-lapse microscopic images (**a**) of HaCat cells infected by *C. albicans* in the presence of different concentrations of MICAN and the corresponding hyphal growth curves (**b**) determined from the average individual hyphal area of *C. albicans*. (Areas were measured instead of lengths due to the hyphal ramifications).

**Figure 2 molecules-25-00903-f002:**
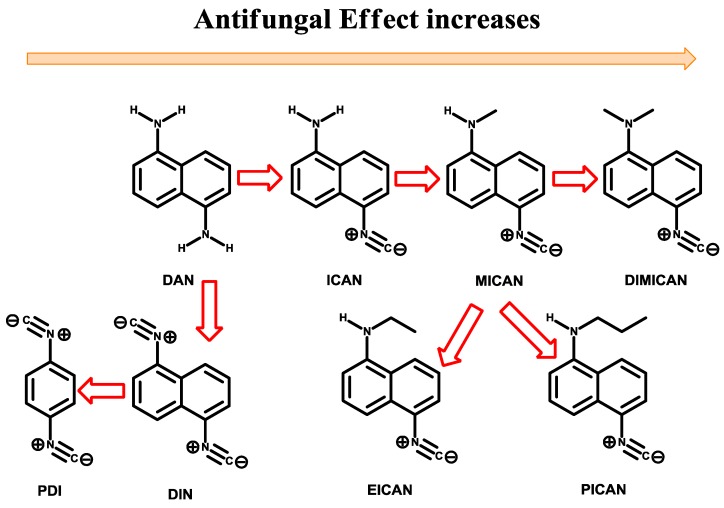
The development strategy for the most effective 1-amino-5-isocyanonaphthalene (ICAN) derivative. For the full names please refer to the experimental section of this paper.

**Figure 3 molecules-25-00903-f003:**
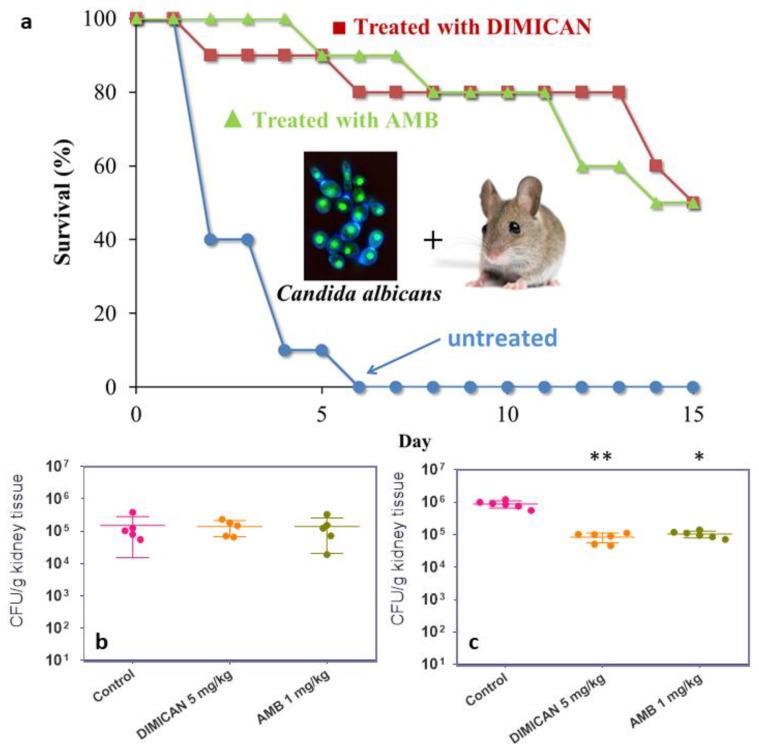
In vivo antifungal activity results for DIMICAN against *Candida albicans* 3666 isolate. (**a**) Survival curves of mice infected with *Candida albicans* 3666 isolate. For all groups *n* = 10. (**b**) Kidney tissue burden of immunosuppressed mice infected by *Candida albicans* 3666 isolate, two days after infection. (**c**) Six days after infection (*p* < 0.05 (*); *p* <0.001 (**)).

**Table 1 molecules-25-00903-t001:** In vitro susceptibilities of *C. albicans* SC5314, *C. parapsilosis* ATCC 22019, and *C. krusei* ATCC 6258 to various ICAN derivatives determined by the Clinical & Laboratory Standards Institute (CLSI) broth microdilution method. MICs were read using the partial (MIC_PI_) and total inhibition (MIC_TI_) criteria. [22]. For the full names and structures of the compounds please refer to Figure 2 and the experimental section of this paper. The lowest MICs are indicated in red. For the MIC_TI_ values of AMB for these *Candida* strains please refer to Table 2.

Compound	*Candida* Species	MIC_PI_ (µg/mL)	MIC_TI_ (µg/mL)
***ICAN***	*C. albicans* SC5314	1.25	5
	*C. parapsilosis* ATCC 22019	0.3	0.6
	*C. krusei* ATCC 6258	0.6	2.5
***MICAN***	*C. albicans* SC5314	0.3	1.25
	*C. parapsilosis* ATCC 22019	0.15	0.75
	*C. krusei* ATCC 6258	0.15	1.25
***EICAN***	*C. albicans* SC5314	0.6	2.5
	*C. parapsilosis* ATCC 22019	0.6	1.25
	*C. krusei* ATCC 6258	0.6	1.25
***PICAN***	*C. albicans* SC5314	0.6	2.5
	*C. parapsilosis* ATCC 22019	0.6	1.75
	*C. krusei* ATCC 6258	0.15	2.5
***DIMICAN***	*C. albicans* SC5314	**≤0.04**	**0.6**
	*C. parapsilosis* ATCC 22019	**0.15**	**0.6**
	*C. krusei* ATCC 6258	**0.15**	**0.3**
***DIN***	*C. albicans* SC5314	0.3	0.6
	*C. parapsilosis* ATCC 22019	0.3	0.6
	*C. krusei* ATCC 6258	0.3	0.6
***DAN***	*C. albicans* SC5314	50	100
	*C. parapsilosis* ATCC 22019	50	100
	*C. krusei* ATCC 6258	50	100
***PDI***	*C. albicans* SC5314	50	100
	*C. parapsilosis* ATCC 22019	50	100
	*C. krusei* ATCC 6258	50	100

**Table 2 molecules-25-00903-t002:** In vitro susceptibilities of *Candida isolates* to DIMICAN and amphotericin B (AMB), determined by CLSI broth microdilution method. MICs were determined according to the CLSI M27-A3 using the partial (MIC_PI_) and total inhibition (MIC_TI_) criteria. [22]. MICs were read visually after 48 h incubation at 35 °C using the partial (50%) inhibition (for DIMICAN) (PI) and total inhibition (for AMB and DIMICAN) (TI) criteria. MICs were determined at least twice. DIMICAN = 1,1-*N*-dimethylamino-5-isocyanonaphthalene.

*C. albicans*	*C. tropicalis*	*C. krusei*
Isolate	MIC_PI_ (µg/mL)	MIC_TI_ (µg/mL)	AMB MIC_TI_ (µg/mL)	Isolate	MIC_PI_ (µg/mL)	MIC_TI_ (µg/mL)	AMB MIC_TI_ (µg/mL)	Isolate	MIC_PI_ (µg/mL)	MIC_TI_ (µg/mL)	AMB MIC_TI_ (µg/mL)
ATCC 10231	≤0.04	0.3	0.25	ATCC 750	0.15	1.25	0.5	ATCC 6258	0.15	0.3	1
456	≤0.04	0.6	0.5	579	0.08	0.6	0.5	749	0.08	0.08	1
665	≤0.04	0.6	1	622	0.08	0.3	1	1853	0.08	0.3	1
685	≤0.04	0.6	0.25	1776	0.08	0.3	1	4364	≤0.04	0.15	1
720	≤0.04	0.6	0.5	2467	0.15	1.25	0.5	5029	≤0.04	0.08	1
963	≤0.04	0.6	0.5	3014	0.08	0.6	0.5	5072	0.08	0.3	1
1544	≤0.04	0.6	0.25	5115	0.08	1.25	0.5	25504	0.08	0.3	0.5
8658	≤0.04	0.6	1	5806	0.08	0.15	0.25	34987	≤0.04	0.08	1
3666	≤0.04	0.6	0.5								
***C. glabrata***	***C. parapsilosis***				
Isolate	MIC_PI_ (µg/mL)	MIC_TI_ (µg/mL)	AMB MIC_TI_ (µg/mL)	Isolate	MIC_PI_ (µg/mL)	MIC_TI_ (µg/mL)	AMB MIC_TI_ (µg/mL)				
ATCC 90030	0.15	0.3	0.5	ATCC 22019	0.15	0.6	0.5				
14408	≤0.04	0.08	0.5	4133	0.08	0.08	0.25				
14559	≤0.04	0.3	0.5	9613	0.15	0.15	0.25				
22394	≤0.04	0.3	0.5	10252	≤0.04	0.08	0.5				
23429	≤0.04	0.3	1	18154	0.08	0.6	0.5				
23481	0.08	1.25	1	27001	≤0.04	0.3	0.25				
27643	0.08	0.3	0.5								
28557	≤0.04	0.6	0.5

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
