# Peer review of "Antifungal Activity of an Original Amino-Isocyanonaphthalene (ICAN) Compound Family: Promising Broad Spectrum Antifungals"

_molecules, 2020, doi:10.3390/molecules25040903_

Round 1

Reviewer 1 Report

The manuscript entitled “Antifungal activity of an original amino- isocyanonaphthalene (ICAN) compound family: A promising solution against fungal ‘superbugs’?” presented by Miklós Nagy and co-workers, evaluates the effect of 1-amino-5-isocyanonaphthalenes against different clinically relevant Candida species. I found the manuscript really interesting, but there are two vital points that raise my concern:

Major issues:

The authors said that they used ICAN as a vital dye on HaCat. Then, on the results section, the authors say that ICAN and its derivatives “are well tolerated by human cell cultures”. The authors need to further explore this issue. As I understood, the LD50 was determined for HaCat, which are cells of the epidermis, and the authors are claiming the potential of the molecules for generalized fungal infections, what was justified by their administration through intraperitoneal injection in the mouse models. It is mandatory that the authors add some information regarding the toxicity of the compounds in other human cell lines, such as hepatic, gastroenteric and endothelial. This information is necessary and will significantly strength the importance of their results. Did the authors injected the vehicle of the formulation (olive oil/water, with no DIMICAN) in infected and non-infected mouse, for comparison? This is determinant to determine the effect of the vehicle.

Minor issues

I suggest the authors to make a careful English revision. There is a lack of prepositions and punctuation throughout the manuscript that, in most cases, makes the sentences confuse. Lines 50-51: Please rewrite the sentence “However, they already found numerous and versatile use in both analytical chemistry [15-17] and cell-biology [18-20].”; its linkage with the previous sentence is not very clear. Table 2: The information of the table is not very clear. I suggest the authors to add table footnotes to better explain it. For instance, MICPI and MICTI represent the partial and total inhibition for DIMICAN? Only the total inhibition value is presented for AMB? What percentage of inhibition does the MICPI refer to? Figure 3: Through the analysis of the figure, it can be seen that there is about 50% mortality on day 15 after infection. Do the authors have some comment about this, for instance, related to the possible toxicity of both drugs? Supplementary Data: Please check the movies Labels. It seems that they are with wrong order, since DIMICAN shows hyphae growth, while MICAN shows an almost complete growth inhibition, and it was not supposed to, once DIMICAN is more active. Are ICAN derivatives only fungistatic or also fungicidal? Do the authors have any information about this? The discussion section is a bit poorly explored. Please improve.

Author Response

11 February, 2020

Dear Reviewer,

Thank you for reviewing our manuscript. Changes have been made in the manuscript based on your comments that are listed below. The changes in the manuscript are highlighted in yellow.

Your comments are presented in italic style text.

Major issues:

The authors said that they used ICAN as a vital dye on HaCat. Then, on the results section, the authors say that ICAN and its derivatives “are well tolerated by human cell cultures”. The authors need to further explore this issue. As I understood, the LD50 was determined for HaCat, which are cells of the epidermis, and the authors are claiming the potential of the molecules for generalized fungal infections, what was justified by their administration through intraperitoneal injection in the mouse models. It is mandatory that the authors add some information regarding the toxicity of the compounds in other human cell lines, such as hepatic, gastroenteric and endothelial. This information is necessary and will significantly strength the importance of their results.

We completely agree with your valuable comment. The test of the toxicity of the ICAN derivatives in other human cell lines, such as hepatic, gastroenteric and endothelial would, indeed strenghten the manuscript. However, it should be noted that this in only the first step in a series of investigations and we only wanted to show the antifungal activity of our ICAN derivatives and their potential to become an original drug family.

In our previous studies we have shown, that the ICAN derivatives can be used as vital cell stains, not only on HaCat, but in other human cell lines such as CaCo2, OCM-1, HuLi, too. Please see Ref. 20.

Z. Nagy, M. Nagy, A. Kiss, D. Rácz, B. Barna, P. Könczöl, C. Bankó, Z. Bacsó, S. Kéki, G. Banfalvi, G. Szemán-Nagy, MICAN, a new fluorophore for vital and non-vital staining of human cells. Toxicology in Vitro 48, 137-145 (2018).

We have added these cell lines to the Introduction in lines 51-53 as:

„During our experiments to utilize ICAN derivatives as vital stains on CaCo2, OCM-1, HuLi and HaCat cell lines [20] we noticed that they are perfectly suitable for the staining of different fungi too.”

Based on these promising previous results we assumed, that they possess low toxicity in human/animal cells, therefore we moved to animal tests, which turned out to be also successful. Nevertheless, in the long run it is important to determine the toxicity of ICANs in various human cell lines, but we think this is the topic of another paper.

Did the authors injected the vehicle of the formulation (olive oil/water, with no DIMICAN) in infected and non-infected mouse, for comparison? This is determinant to determine the effect of the vehicle.

Yes we tested the emulsion alone in 8 week old mice and the results are summarized in the following Table.

Mice, (both male and female) treated with the olive oil/water emulsion showed an increased weight gain (approximately 10%) compared to that of the control group. No mortality due to the emulsion was observed within the 5 days treatment.

Minor issues

I suggest the authors to make a careful English revision. There is a lack of prepositions and punctuation throughout the manuscript that, in most cases, makes the sentences confuse. Lines 50-51: Please rewrite the sentence “However, they already found numerous and versatile use in both analytical chemistry [15-17] and cell-biology [18-20].”; its linkage with the previous sentence is not very clear.

The beginning of the sentence has been rephrased as:

Despite their relative novelty, they already found numerous and versatile use in both analytical chemistry [15-17] and cell-biology [18-20].

Table 2: The information of the table is not very clear. I suggest the authors to add table footnotes to better explain it. For instance, MICPI and MICTI represent the partial and total inhibition for DIMICAN? Only the total inhibition value is presented for AMB? What percentage of inhibition does the MICPI refer to?

This information is already given in the original text in lines 104-107:

“MICs were read visually after 48-h incubation at 35 oC using the partial (50%) inhibition (for ICAN derivatives) (PI) and total inhibition (for AMB and ICAN derivatives) (TI) criteria. MICs were determined at least twice.”

However, the following text was added to the caption of Table 2:

„MICs were read visually after 48-h incubation at 35 oC using the partial (50%) inhibition (for DIMICAN) (PI) and total inhibition (for AMB and DIMICAN) (TI) criteria. MICs were determined at least twice.”

Figure 3: Through the analysis of the figure, it can be seen that there is about 50% mortality on day 15 after infection. Do the authors have some comment about this, for instance, related to the possible toxicity of both drugs?

We don’t know what causes the 50% mortality. Since this is the very beginning of drug development it can be attributed to many factors, the determination of which requires further studies.

Supplementary Data: Please check the movies Labels. It seems that they are with wrong order, since DIMICAN shows hyphae growth, while MICAN shows an almost complete growth inhibition, and it was not supposed to, once DIMICAN is more active.

The labels are OK. Please note that the concentrations of DIMICAN are one order of magnitude lower than those of MICAN. These movies were recorded during the preliminary experiments, that is why there is a difference in concentrations.

Are ICAN derivatives only fungistatic or also fungicidal? Do the authors have any information about this?

We have no information about this question. We did not carry out minimal fungicidal concentration or time-kill study tests to determine whether these compounds are fungistatic or fungicidal.

The discussion section is a bit poorly explored. Please improve.  

The discussion section has been improved.

Yours sincerely,         

                                                                                                              Sándor Kéki

                                                                                                          professor and head

Reviewer 2 Report

This manuscript describes experiments that test the antifungal activity of isocyanonaphthalene (ICAN) compounds. The antifungal activity of 1,5-diaminonphthalene (DAN) and its derivatives against a number of strains of the fungal species Candida albicans, Candida tropicalis, Candida krusei, Candida glabrata and Candida parapsilosis was measured with in vitro liquid MIC assays. The activity of the most potent derivative, with the most favourable physico-chemical properties, was tested in vivo using a mouse model of systemic candidiasis. The antifungal activities of the compounds were compared with those of Amphotericin B, and the most potent derivatives showed comparable activity in both the in vitro and in vivo assays. A strength of the study is the large number of Candida strains used in the MIC assays, it is a pity that the compounds were not tested against azole resistant Candida albicans strains or Candida auris.

Major comments

There needs to be a better explanation of the growth inhibitory effect of MICAN (Figure 1b). The methods (lines 236-239) describe image analysis, but there is no description of the co-culture of C. albicans with HaCat cells or how the ‘Hyphal area’ in Figure 1 b was measured. On lines 82-85 it says that MICAN resulted in 30% growth. What does that mean? The units for the y-axis in Figure 1b are µm2. If maximum growth is 120 µm2 (DMSO) the growth in the presence of MICAN looks to be ~10 µm2 which is 8.3% maximum growth. Also, on lines 84-85 it says that the growth in the absence of MICAN was 1000%. What does this mean? The title of the manuscript refers to ‘superbugs’. Superbugs are, by definition, multidrug-resistant. C. albicans most often develops resistance to azoles, and C. krusei and C. glabrata often show reduced susceptibility to azoles. Resistance to amphotericin B is relatively rare. The most serious ‘superbug’ amongst the Candida species is Candida auris which can become resistant to all three classes of antifungal. It would have been good to test the activity of the ICAN compounds against C. auris. If strains of C. auris are not available, it would have been good to measure the strains susceptibility to an azole such as fluconazole (FLC), as well as amphotericin B, and test the compounds against some strains that were FLC resistant. As the study does not test the compounds against ‘superbugs’, and indeed according to the data presented in the manuscript it was not tested against resistant fungal strains, I think that “A promising solution against fungal ‘superbugs’” should be replaced with ‘Promising broad spectrum antifungals’. On lines 104-105 and 250 it says that MICs were determined at least twice. Did the values vary? If so, and if there were two determinations which varied, what is reported in the manuscript? Line 119, I think that DIN has a greater antifungal effect than ICAN as the MIC values for DIN are lower, rather than a lower antifungal effect as stated on line 119. The reason why ICAN was used in subsequent experiments, rather than the better antifungal DIN, should be stated clearly in the results and discussion sections. Figure legends. These should start with a description of the type of data presented and then indicate what is in each panel. For Figures 1 and 3 the legend title (in bold) only refers to panel (a).

Minor comments

The English needs to be checked throughout the manuscript. Lines 19 and 94, I think “structure-effect” should be ‘structure-activity’. Line 23, replace “deeply” with ‘severely’. Lines 33-34, I think Aspergillus is the major cause of invasive fungal infections. Lines 44-45, replace “and few new drugs to come” with ‘and there are few new drugs in the pipeline’. Line 50, I am unfamiliar with the phrase: “white spot on the map”. Lines 101-102, these details are already in the Methods section. Line 67, “As we mentioned in the Introduction chapter” is unnecessary, start the sentence with ‘We previously carried …”. Line 72, replace “on these results” with ‘by these results’. Line 74, replace “fungi species” with ‘fungal species’. Line 87, replace “growth hindering” with ‘growth inhibitory’. Table 2, why are two of the columns coloured yellow/green? Lines 160-161, replace “well seen” with ‘evident’. Figure 3a – I don’t think this needs drawings of live and dead mice, most scientists will understand what ‘survival’ means’. The Materials and Methods section should list the fungal species, their sources and culture conditions.

Reviewer 3 Report

The manuscript presents the results of a study concerning the antifungal effect of a new class of compounds, namely 1-amino-5-isocyanonaphthalenes (ICANs). The paper is well written and the methodology is exhaustively described.

I recommend the change of the title by removing its second part (i.e. "A promising solution against fungal ‘superbugs’?") as Candida species tested in this study are not "fungal superbugs".

It would be interesting to know the effect of this novel antifungal agent against fluconazole-resistant Candida isolates.  

Lines 197-199: The phrase should be rewritten as the toxicity of the compound was not extensively documented and its use as disinfectant must be proved by time-kill assay. 
